# Membrane Separation Processes and Post-Combustion Carbon Capture: State of the Art and Prospects

**DOI:** 10.3390/membranes12090884

**Published:** 2022-09-14

**Authors:** Eric Favre

**Affiliations:** Laboratoire Réactions et Génie des Procédés, CNRS, Université de Lorraine, 54001 Nancy, France; eric.favre@univ-lorraine.fr

**Keywords:** carbon capture, materials, processes, energy

## Abstract

Membrane processes have been investigated for carbon capture for more than four decades. Important efforts have been more recently achieved for the development of advanced materials and, to a lesser extent, on process engineering studies. A state-of-the-art analysis is proposed with a critical comparison to gas absorption technology, which is still considered as the best available technology for this application. The possibilities offered by high-performance membrane materials (zeolites, Carbon Molecular Sieves, Metal Oxide Frameworks, graphenes, facilitated transport membranes, etc.) are discussed in combination to process strategies (multistage design, hybrid processes, energy integration). The future challenges and open questions of membranes for carbon capture are finally proposed.

## 1. Introduction

Global warming issues address huge scientific, technological, economical, and societal challenges. Besides decreased use of fossil fuels, increased energy efficiency, and increased use of renewables, the capture of carbon, more specifically from large emitting sources (power plants, blast furnaces, cement factories, petrorefineries, etc.) is considered as a key lever [1]. After concentration and purification from flue gases, carbon dioxide can either be stored into geological formations (carbon capture and storage, CCS) or transformed into molecules for use as chemicals or for fuel applications (carbon capture and use, CCU).

Generally speaking, the carbon capture step is of major importance because it is the most expensive part of the CCS chain [2]. Moreover, the carbon capture process should require the lowest energy requirement in order to prevent secondary emissions (i.e., CO_2_ produced by the energy used for the capture step). Consequently, very large research efforts have been addressed for decades in order to select and design the most efficient and cost effective carbon capture process. Among the different process possibilities (cryogeny, gas–liquid absorption, adsorption on a solid and membrane separation), the absorption of CO_2_ into a chemically reactive liquid solvent (typically an aqueous amine solution) is considered today as the best available technology [3]. Gas–liquid absorption has indeed been practiced for a long time and on a large scale for natural gas treatment, for which CO_2_ has to be removed from a gas mixture. Solvent regeneration is achieved through steam stripping, leading to a heat duty in the range of 3 GJ per ton of recovered CO_2_ [2,4]. Several demonstrators have been designed and used in order to evaluate the applicability of the gas–liquid absorption technology to capture carbon from flue gases on industrial sites [2].

Membrane processes show promising possibilities for carbon capture but are still considered today as a second-generation technology [4]. This paper intends to achieve a state-of-the-art analysis of membrane processes for carbon capture applications. In a first step, the overall carbon capture framework and the key performance indicators will be detailed. Membrane materials, a very active field of research, will then be summarized before discussing the impact of materials’ performances on process engineering aspects. Future challenges of membranes for carbon capture and some important unsolved issues will finally be analyzed.

## 2. Carbon Capture Framework

The overall framework of carbon capture, be it for CCS or CCU, is typical of industrial separation problems. The solution to these type of problems classically requires to first define the characteristics of the feed mixture (i.e., flue gas composition, flowrate, temperature, and pressure) and the separation process specifications (with CO_2_ purity and recovery being the most important) [2]. Through process simulation and/or pilot experimental data, key performance indicators can then be quantitatively evaluated in order to enable a relevant comparison between different processes or separating agents or operating conditions [5].

As stated above, the specific energy requirement of the capture step (E), usually expressed in GJ per ton of recovered CO_2_, is the indicator number one. For processes making use of energy as power and not heat, such as membrane processes (which require gas compression and/or vacuum pumping), E will logically be expressed in kWh per ton of CO_2_. This is occasionally considered as a drawback of membrane processes, with power being a high-quality energy compared to heat. The need to define a conversion factor between heat duty (Q, for thermal processes) and power requirement (W) can then complicate the comparison. A 2.7 factor (1 J power basis corresponding to 2.7 J heat basis) is occasionally used for that purpose because it corresponds to the average current energy efficiency of power plants, where combustion energy is transformed into electrical power. Nevertheless, a much broader range of efficiency can be found depending on site or country specificities.

Steady-state operating conditions are most often taken into account. Dynamic carbon capture operation, such as intermittent use, fluctuating inlet flowrate, or start and stop aspects, are almost unexplored.

Figure 1 shows a generic carbon capture framework. In terms of specifications, a CO_2_ purity of 90% or more and a capture ratio (CO_2_ recovery) of 90% are classically taken as standards [4,6].

The starting point of the problem clearly corresponds to the flue gas characteristics. The main CO_2_ sources for carbon capture studies are detailed in Table 1, with the corresponding range of inlet CO_2_ content (x_IN_) and major compounds of the mixture. It can be seen that the volume content of CO_2_ to be treated shows a very broad range (in % volume). A very large ratio of carbon capture studies focuses on coal power plants (with ca. 15% in CO_2_). Moreover, most studies limit the analysis to a binary CO_2_/N_2_ mixture, while real flue gases contain water vapor up to saturation level and harmful trace components such as SO_2_ and NOx. More recently, the direct capture of CO_2_ from air (DAC) attracts attention, with an inlet CO_2_ dilution factor of about 300 compared to coal power plant flue gas.

A rigorous analysis of the performances of membrane processes to treat the different gas compositions listed in Table 1 first requires a given membrane material to be selected and the best associated process to be designed. These two objectives are discussed hereafter. The key question is to clearly and critically evaluate the interest of membranes, which are currently used at industrial scale for large applications (air separation, natural gas treatment, hydrogen purification, volatile organic compounds recovery from air, etc. [7,8]) for carbon capture. Interestingly, membrane operation is considered as the best available technology for biogas upgrading, which shows at first glance similarities with flue gas treatment (e.g., elimination of CO_2_ from a gas mixture with a CO_2_-selective membrane). The carbon capture application, however, strongly differs in that the target compound is recovered on the low-pressure (permeate) side for flue gas treatment, thus requiring a recompression step. On the contrary, natural gas (CH_4_), the target compound of biogas upgrading operation, is recovered on the high-pressure (retentate) side.

## 3. Membrane Materials

The search for efficient membrane materials for carbon capture is a very active field, with thousands of different structures reported or patented. Figure 2 tentatively dresses a classification of the different types of materials that can be proposed for carbon capture. Besides the specific case of high permeability materials for membrane contactors (developed for intensified gas absorption processes [9,10]), a large majority of researches look for CO_2_/N_2_ separation application. Basically, the two main materials’ performances, which play a key role in that latter case, correspond to the separation efficiency (typically CO_2_/N_2_ selectivity, dimensionless ratio of CO_2_ over N_2_ permeability) and productivity (membrane permeance, usually expressed in GPU) [7,11]. Dense polymers with a physical separation mechanism (i.e., non-reactive membrane) have been proposed for a long time for that purpose [11,12,13]. A trade-off between permeance and selectivity is observed in that case [14].

The two major performance indicators of membranes for separation purpose are CO_2_/N_2_ selectivity and CO_2_ permeance. From early studies on membranes for post-combustion carbon capture [15,16] to recent, advanced materials produced at lab scale [17,18], a very large number of structures and molecular separation matrices has been reported. A detailed analysis of the different performance levels that have been achieved for each type of material and separation mechanism is beyond the scope of this paper. A simplified synopsis is proposed on Table 2, and a graphical trade-off representation with the different categories of membranes listed in Table 2 is shown on Figure 3.

The improvements that have been more recently obtained, be it thanks to nanostructured materials or chemically reacting systems, is noticeable. For instance, record permeances up to 10,000 GPU have been reported for graphene membranes, while amine-based reacting membranes enable CO_2_/N_2_ selectivity close to 800 to be reached.

This new set of performances logically addresses the question of the impact on process key indicators listed in Figure 1 (especially maximal CO_2_ purity, energy, footprint). The answer to this important question requires process engineering analyses, which are discussed in the next section.

## 4. Process Engineering

### 4.1. Single-Stage Performances and Limits

The starting point of membrane process evaluation for carbon capture consists to explore the possibilities and limits of a single membrane stage with a given membrane material (i.e., fixed selectivity and permeance). The search for an energy requirement level possibly lower than 2 GJ per ton of recovered CO_2_ (which can roughly be estimated to 200 kWh equivalent) is the target number one.

Early attempts in that direction concluded that membranes cannot reach this energy efficiency level when a 90% purity and 90% capture ratio are imposed [15,16]. A systematic parametric analysis concluded later on that vacuum pumping is absolutely necessary in order to decrease the energy requirement [34,35]. Furthermore, a single membrane stage shows a very strong sensitivity towards the inlet CO_2_ content [36]. Figure 4 shows that the target CO_2_ purity (y = 0.90) and energy requirement (i.e., 2 GJ per ton max) can be reached with one stage only when the inlet CO_2_ content is larger than 30%. For coal power plant flue gas (x_IN_ = 0.15), the priority number one of post-combustion carbon capture, a moderate CO_2_ purity can be achieved, and a membrane selectivity (α) above 50 is needed.

The performance analysis sketched in Figure 4 has several implications in terms of practical use of membranes for coal power plant flue gas carbon capture:-Based on the currently available membrane materials (e.g., Polaris in Table 2 [19]), a two-stage (or more) process is needed. Alternatively, a hybrid process combing a membrane concentration step and a cryogenic polishing unit can be proposed (e.g., Air Liquide low-temperature Cryocap process [37]).-Vacuum pumping is usually favored, associated to a moderate feed compression, in order to reach the energy requirement. From an industrial point of view, vacuum operation is most often unwanted, but for carbon capture, this option is almost unavoidable. A moderate vacuum, typically around 10 to 100 mBar, can be operated for large-scale installations based on liquid ring or primary dry pumps, which generate a large footprint area. Lower vacuum levels can be achieved at lab scale, but vacuum pumps energy efficiency often drops for low-pressure levels, and leaks make low pressure very difficult to attain.-Because of the very high sensitivity of inlet CO_2_ content on purity and energy, a strategy of partial exhaust CO_2_ recycling in order to increase inlet CO_2_ content can be of interest (e.g., MTR process [19]).

### 4.2. Multistage Processes

Given the limitations of a single-stage process, a significant number of studies explored the search for the optimal design of multi-staged membrane carbon capture processes [38,39,40,41,42,43,44]. A cost function is necessarily required in that case, with CAPEX- (compressors, vacuum pumps, membrane modules) and OPEX- (electricity) specific costs. A large number of possibilities in terms of process options (vacuum pumping allowed or not, partial recycling loops or not, different pressure ratio per stage or not, energy recovery system or not, dry or wet flue gas, etc.), specifications (purity and recovery), and optimization methods can be found. As a consequence, the optimal process design for each type of situation is not clearly established yet. The general trends of the different process design studies can be summarized as follows:-Two-stage designs, including one or two recycling loops, are usually favored for sake of simplicity (Figure 5). It is interesting to note that the most frequent design of simulation/optimization studies for a coal power plant flue gas, shown on Figure 5b, is similar to the structure of natural gas or biogas upgrading membrane processes [11]. This solution is typical of a purity/recovery constraint when membranes are used [13].-Vacuum pumping is most often applied, and it is useful for energy requirement decrease constraints, but this translates into low driving forces, that is, a very large membrane surface area.-Very high membrane permeance levels are needed (mostly due to the previous item).-The impact of water is most often neglected (a dry inlet mixture is assumed), but it can affect the set of performances [45]. More specifically, chemically reactive membranes require a humid gas in order to enable the chemical reaction to take place on both sides of the membrane [46]. The resulting process simulation problem is tricky (variable permeability, correct computation of water fluxes).-The same membrane type (i.e., same selectivity and permeance) is used for the different stages except in very few studies.-Membrane pre-treatment operation, such as dust or SOx/NOx removal, are not considered in the analysis.-Besides membrane processes, hybrid designs combining a membrane CO_2_ preconcentration step and a polishing step (often based on cryogeny) are usually considered as the more relevant [8,36]. Two examples are shown on Figure 6.

Globally speaking, the process structures shown on Figure 5 and Figure 6 are quite simple, and it could be expected that clear and systematic conclusion can be drawn. This is not the case, and there is no consensus, especially on membrane and overall carbon capture cost (which can vary from USD 25 to USD 500 per ton of recovered CO_2_). Any decisive comparison to gas–liquid absorption, the best available technology with specific costs in the range of USD 50 to USD 80, is then difficult. It is also important to point out that costs widely vary from country to country as a function of time. Using power and duty requirements could thus be a fairer comparison.

Besides process simulation studies, a new strategy, namely process synthesis, can also be proposed. The key concept is to systematically explore the different connection possibilities through superstructure, neural networks, or genetic algorithms in order to identify, among the whole set of architectures, the optimal process design and associated operating conditions [47]. Interestingly, the advantage of advanced materials or of combining different materials in multistage systems can be rigorously analyzed thanks to these recent tools [48]. Figure 7 shows an example of the key framework of membrane process synthesis studies, i.e., the different superstructures that can be generated for a three-stage membrane process including vacuum pumping, partial recycling loops, and energy recovery systems. The exhaustive exploration of the different design options taking into account different flue gas contents, different membrane performances, and cost is not fully achieved yet. Moreover, different specific cost functions are used, with a large range of membrane and compressor costs. This logically impacts the optimal design. More specifically, vacuum pumping operation or partial recycling loops are most often not included in the optimization packages [49]. It is expected, however, that advanced computing and process synthesis methods making use of artificial intelligence tools will quickly provide robust answers on the most efficient process for a given membrane set of performances and given flue gas content.

## 5. Open Questions, Further Research, and Prospects

A synthetic state of the art on research in the field of membrane materials and processes for carbon capture is proposed in the previous sections. Impressive efforts and progress have been achieved, but it is absolutely necessary to go further so that membrane is considered as a liable carbon capture process (i.e., a high TRL process). From a practical point of view, for instance, a very limited number of pilot installations based on tailor-made modules with dense polymeric membranes has been operated on real flue gases [50]. This matter of fact can be considered as one the main limitations of membranes in the carbon capture field. In the last part of this paper, open questions and prospective approaches are discussed in order to further promote the interest of membranes for carbon capture.

### 5.1. Materials

The search for constantly improved membrane performances, especially for CO_2_/N_2_ separation, is still very active in the materials community. Impressive progresses have been achieved, probably opening new perspectives (that remain to be clearly established). It should be noted, however, that a large ratio of publications is limited to pure gases results or model dry CO_2_/N_2_ mixtures. Experiments with complex mixtures in order to approach the real flue gas case should clearly be more systematically undertaken [51].

In a very large number of cases, CO_2_ permeates faster than N_2_ in membranes. This particularly applies for polymers (unless unsteady operation in a chemically reacting membrane is achieved) and nanostructured materials. The possibility to develop a reverse selective behavior is extremely rare but has been reported (with metallic membranes) [52,53]. This peculiarity allows CO_2_ to be recovered on the high-pressure outlet (retentate), which is favorable for further compression operations (typically up to 110 bar for CCS). The interest of reverse selective membranes should be more systematically investigated together with the interest to combine CO_2_ selective and N_2_ selective materials in multi-staged processes. The early attempts in that direction should be more systematically developed both from the materials and process design point of views.

Another open question concerns high-temperature separations, which are enabled by ceramic or metallic membranes. The joint need to use a high-temperature (>200 °C) energy input with a steam sweep operation requires a complete reassessment of process implementation in a carbon capture context. Process optimization studies (with effective chemically reactive membrane simulation packages) are needed in order to evaluate the energy requirement and overall capture cost of this new type of high-performance materials. It might be that a combination of high-temperature chemically reactive membranes with low-temperature physical separation membranes offers attracting performances in some cases.

Coming back to the comparison to absorption processes, it is interesting to notice that membrane development somehow rediscovered the key molecular systems operated by gas–liquid absorption units for natural gas treatment for decades (Table 3). Basically, low-temperature physical separation membranes make use of glycol-type moieties in block copolymers. This solution has been soon identified for CO_2_ capture with physical absorption solvents. Similarly, amine-based systems, largely applied by natural gas treatment companies, are the most common solution for low-temperature chemically reactive membranes. Finally, the high-temperature carbonate membranes use the same molecular system as Benfield hot carbonate process [54]. This set of observation is by no means exhaustive. For instance, sterically based membrane materials (such as nanostructured) offer unique separation mechanisms and performances compared to solvents. Nevertheless, the similarities summarized in Table 3 may offer interesting cross-comparison studies between absorption and membrane processes.

### 5.2. Processes

It has been stated above that, comparatively to materials’ efforts, process design studies dedicated to carbon capture by membranes are more limited. The general simulation framework of membrane gas separations is well-established today, based on the pioneering developments of Weller and Steiner [55]. Constant permeability hypothesis usually provides efficient predictions of module performances [56] and corresponds to the cornerstone of current process optimization investigations. Generally speaking, two-stage systems making use of a vacuum are proposed based on process simulation studies in order to fulfill the purity, recovery, and energy requirement constraints [50].

A different extension to this classical framework can be useful. Variable permeability or chemically reactive systems can be correctly simulated today [46,57] but are not often considered in process optimization studies. Pressure-drop effects are also important but are rarely considered in simulation studies because they are geometry- (module) and gas-velocity-dependent. Moreover, the high selectivity and permeability that is attainable by advanced materials will for sure generate concentration polarization effects (i.e., mass transfer resistance in the gas boundary layer) [58,59]. This important limitation has not been taken into account by process synthesis approaches. Efforts should be made in this direction in order to achieve a correct evaluation of advanced membrane processes for carbon capture.

Given the key importance of the driving force aspects, (which govern the process energy requirement to a large extent), alternative driving forces such as sweep operation, temperature gradients, or electro-swing approaches [60] should be more systematically investigated. The minimum entropy dissipation concept [61] could possibly help in order to identify the best set of driving forces for carbon capture, but this strategy remains unexplored.

Besides energy requirement, another KPI, namely the carbon capture unit footprint, should also be taken into account (Figure 1). Very few evaluations are reported on this indicator, which should strongly favor membranes vs. absorption units. Membranes are known indeed to enable significant process intensification, especially due to the high specific surface area of modules. For a quantitative comparison, a classical 1 mol CO_2_ m^−3^·s^−1^ can be taken for the average steady-state volume absorption capacity of gas–liquid systems for post-combustion [9]. Taking into account the additional stripping column and overall solvent loop equipment, the key advantage of membrane process (no regeneration step needed) surely translates into a large-volume reduction that is not pointed out enough.

In terms of process synthesis tools, two important open questions (among others) can be mentioned. First, the optimal process design of very high-selectivity/very high-permeance membranes is not clearly addressed so far. Beyond a 200 CO_2_/N_2_ selectivity threshold, a single-stage membrane process can indeed achieve the 90% purity/recovery target (Figure 4). The problem of high-selectivity membranes is, however, that driving force quickly vanishes, leading to unacceptable membrane surfaces [7,13]. A potential solution in that case is to achieve a partial retentate-to-permeate recycle, such as systematically applied in membrane gas-drying operation [13]. The validity of this prospective analysis, sketched out in Figure 8a, remains to be formally proven by a rigorous process synthesis study with high-performance materials (such as the group #4 in Figure 3).

A second important process design problem consists of investigating a multi-stage membrane process including all options in terms of materials (multimembrane system), driving force (compression, vacuum, sweep), and overall structure (partial recycling loops). An example is shown in Figure 8b. Again, no study offering these set of options is reported today. It would be extremely interesting to analyze the outcome of such a study, especially the open questions that remain on the best set of membrane type and performances for carbon capture, such as high-permeance vs. high-selectivity material, physical vs. chemical separation process, multi-membrane vs. single-membrane systems, and high-temperature vs. low-temperature process.

Finally, it is useful to take into account the fact that membrane’s best role for carbon capture is to provide an energy efficient pre-concentration function [36]. Hybrid processes are then often favored in order to achieve the purity/recovery challenge [7,62,63]. Several studies addressed the simulation of membrane-cryogeny systems, but other possibilities such as membrane-absorption [64] or membrane-adsorption are essentially unexplored. The latter could be of interest for the direct air capture (DAC) problem [65]. The development of rigorous process synthesis packages including membrane and absorption/adsorption units is of course an essential prerequisite in order to obtain answers in the interest (or not) of hybrid processes for carbon capture.

## 6. Conclusions

Membrane processes as a possible option for carbon capture have shown a significant change in four decades, from an inadequate technology status to promising second-generation solution. The joined, synergistic developments on tailor-made membrane materials and process design studies enabled this evolution.

Numerous challenges and unsolved issues remain, but it can be anticipated that membrane processes will for sure play a role in the overall carbon capture framework in the near future either as a standalone system or through hybrid processes. Besides a close collaboration between materials science and process optimization, pilot plants operated on real flue gases are, to that respect, an absolute necessity to make membrane carbon capture a reality.

## Figures and Tables

**Figure 1 membranes-12-00884-f001:**
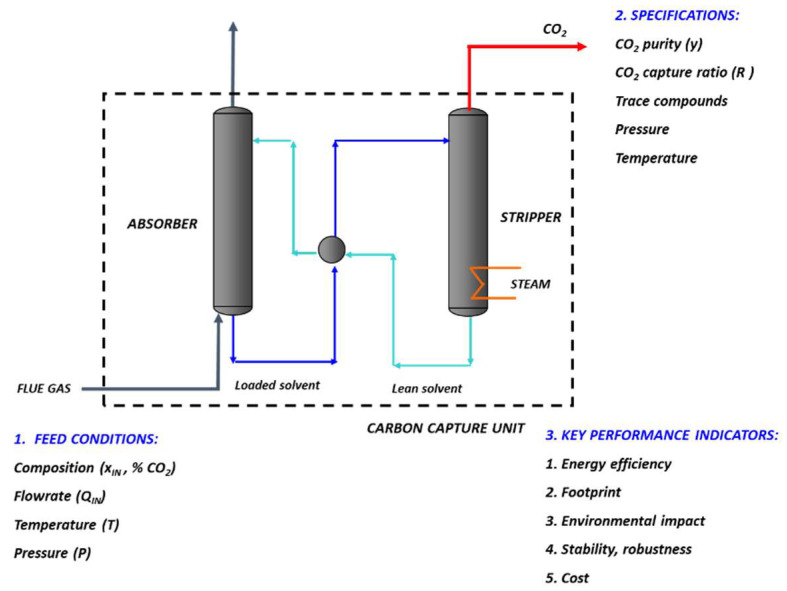
Overall framework of a gas–liquid absorption carbon capture process (baseline technology). Based on inlet flue gas conditions and operating conditions, outlet CO_2_ specifications can be evaluated, and process key performances indicators (KPI) can be assessed. The generic target of membrane processes for carbon capture is to replace the capture unit (dotted line).

**Figure 2 membranes-12-00884-f002:**
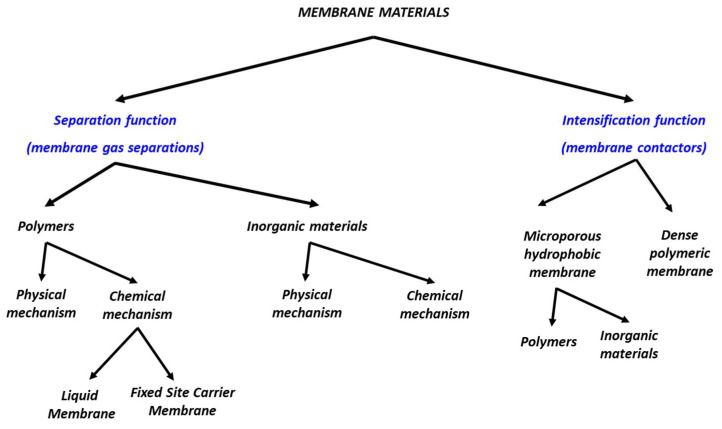
Synopsis of the different types of membrane materials for carbon capture application.

**Figure 3 membranes-12-00884-f003:**
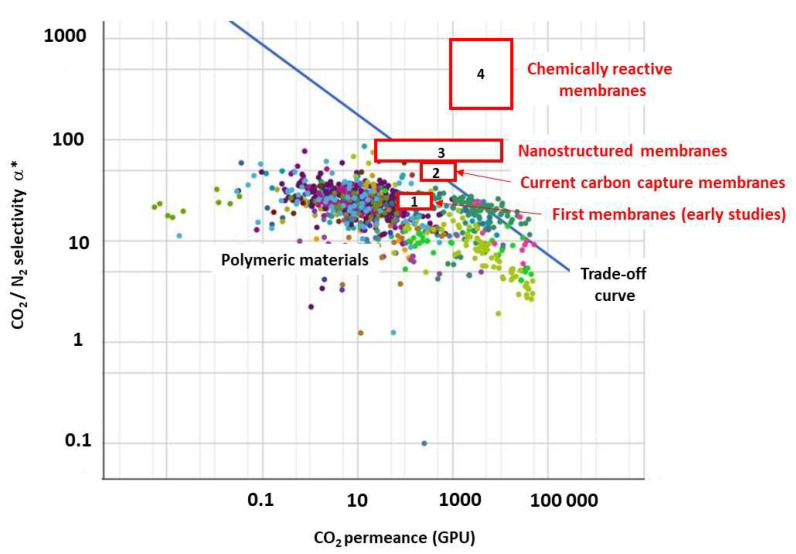
Synthetic representation of the different performances of membrane materials for CO_2_/N_2_ separation (selectivity vs. permeance). Dots correspond to polymeric materials, with the associated trade-off curve. Numbers refer to the performance range of the materials detailed in Table 2. 1, early studies; 2, current commercially available materials; 3, inorganic nanostructured materials; 4, chemically reactive membranes.

**Figure 4 membranes-12-00884-f004:**
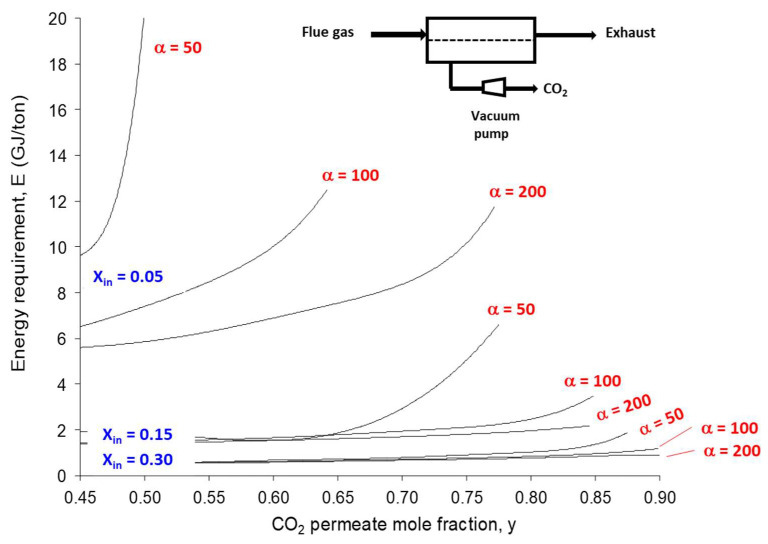
Overall framework of membrane carbon capture process with a single stage: a 90% capture ratio is fixed, and CO_2_ purity (y) and minimal energy requirement is computed for different inlet CO_2_ content (x_IN_) and membrane selectivity (α) values. Vacuum pumping is applied in order to achieve the lowest energy requirement. A 2.7 factor is used to convert membrane unit power requirement to GJ on a thermal basis (y axis).

**Figure 5 membranes-12-00884-f005:**
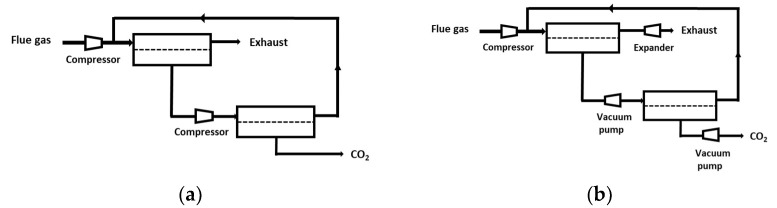
Classical two-stage membrane carbon capture processes: early design (**a**); typical framework of current technico-economic studies (**b**) [38,39,40,41,42,43,44].

**Figure 6 membranes-12-00884-f006:**
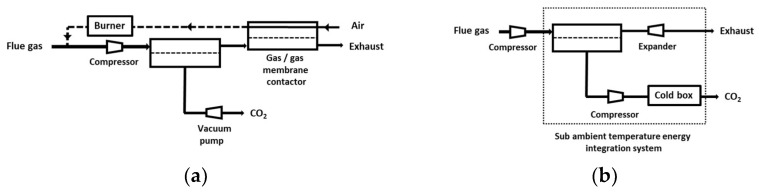
Two examples of hybrid membrane/cryogeny carbon capture processes. MTR process, including vacuum pumping and a retentate recycling concept (in order to increase CO_2_ inlet content), (**a**) [19] and Air Liquide integrated low-temperature membrane stage process, including compression and energy recovery with a cryogenic step (**b**) [37].

**Figure 7 membranes-12-00884-f007:**
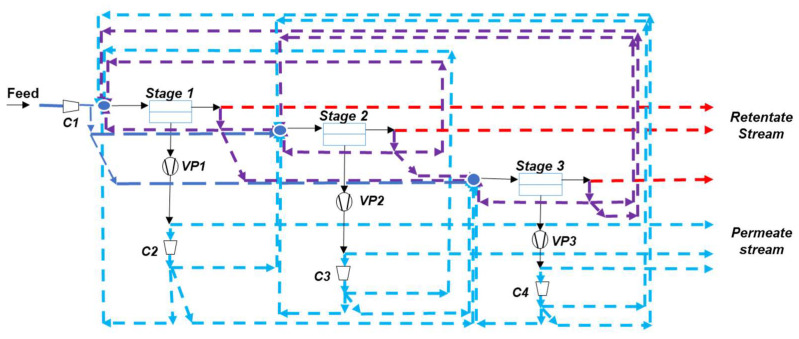
Generic design possibilities for a three-stage membrane process including vacuum pumping, inlet feed compression, and intermediate partial or integral recirculation loops. C stands for compressor and VP for vacuum pump. From [47], adapted.

**Figure 8 membranes-12-00884-f008:**
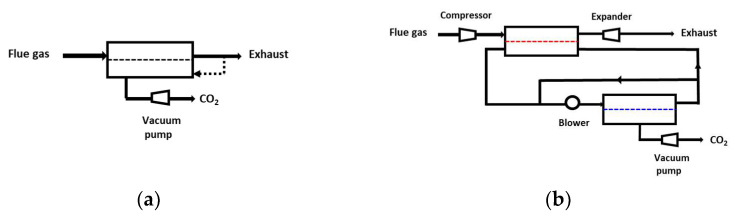
Prospective future membrane capture processes: (**a**) single-stage process based on very high performance material. Similarly to membrane gas-drying operation, retentate recycling could be applied in order to boost driving force. (**b**) A multi-membrane, multi-pressure ratio carbon capture process including a complete set of options (e.g., sweep mode, partial recycling loops) for searching the optimal carbon capture solution.

**Table 1 membranes-12-00884-t001:** Overview of the main sources of carbon dioxide for carbon capture processes.

Source	% CO_2_ (x_IN_)	Other Compounds
Power plant (coal)	12–15	N_2_ (O_2_)
Power plant (gas)	4–5	N_2_ (O_2_)
Steel	5–20	CO, N_2_, H_2_
Cement	20–30	N_2_ (O_2_)
Petrochemicals	10–30	N_2_
Waste incineration	5–15	N_2_ (O_2_)
Biomass boilers	5–15	N_2_ (O_2_)
Biogas	40–60	CH_4_
Air (DAC)	4 × 10^−4^	N_2_, O_2_

**Table 2 membranes-12-00884-t002:** Synopsis of the different materials performance for carbon capture application.

Material Type	CO_2_/N_2_ Selectivity (–)	CO_2_ Permeance (GPU *)	Reference
*Early studies (dense polymers)*			
PPO	19	375	[15,16]
PI	43	100	
*Current commercially available polymeric membranes*			
Polaris (MTR)	50	2200	[19]
Polyactive (Hereon)	46	1450	[20]
*High-performance non-commercially available materials*			
Polymers	78	3000	[18]
Silica	50	900	[21]
Zeolite	69–170	2100–4000	[22,23,24]
Carbon molecular sieve	25	9000	[25]
Graphene	30	10 000	[26]
*Chemical separation mechanism*			
Fixed site carrier	165	1450	[27,28]
Liquid membrane (LM)	140	3000	[29,30]
Enzymatic LM	788	2600	[31]
Hydroxide ceramic **	1000	250	[32,33]

* 1 GPU = 10^−6^ Ncm^3^·cm^−2^·s^−1^·cmHg^−1^. ** At 250 C with steam sweep.

**Table 3 membranes-12-00884-t003:** Tentative analogy between gas–liquid absorption and membrane processes for carbon capture application.

Molecular Mechanism	Temperature Range	Absorption Process	Membrane Process
Physical	−10–60 °C	Ethylene glycol based solvents (Selexol)	Ethyleneglycol-based dense polymers
Chemical	40–120 °C	Amine solvents	Amine-based reacting membranes FSCM, LM
High-temperature chemical	>200 °C	Hot carbonate	High-temperature carbonate-based membranes

## Data Availability

Not applicable.

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
