# Peer review of "Membrane Separation Processes and Post-Combustion Carbon Capture: State of the Art and Prospects"

_membranes, 2022, doi:10.3390/membranes12090884_

Round 1

Reviewer 1 Report

The contribution Membrane processes and carbon capture: State of the art & prospects gives a very good, comprehensive and at the same time concise overview on the use of membrane technology for carbon capture. It highlights the important aspects and most importantly gives good hints for the direction of future research. There are a few minor issues, that should be addressed:

Page 1, line 12: Facilitated transport membranes should also be mentioned as an example

Page 2, lines 62-64: This should be explained more clearly. Heat energy is worth less than mechanical energy?

Page 3, lines 78-84: It should be mentioned already here (later it is discussed) that generally water vapour up to saturation level is present and that various, potentially harmful trace components SO2, NOX should be considered when designing a unit.

Page 3, Table 1, “Steel row”: comma missing between H2 and N2

Page 5, line 134: Was this number calculated using the 2.7 factor mentioned above?

Page 5, line 154: The large footprint area that would result when operating vacuum machinery at pressures of 10  to 100 mbar should be mentioned.

Page 7, line 195: please give a reference for these studies.

Page 7, Figure 5: (b) is missing in the figure caption.

Page 7, lines 209-214: One remark is that costs widely vary from country to country and even within one country as a function of time - just consider the current situation with Russia's invasion of the Ukraine and the resulting turmoil in the energy markets - maybe using power and duty requirements as outlined above is the fairer comparison.

Page 10, lines 297-305: Concentration polarisation an pressure drops as well as different membrane module designs were e.g. considered in International Journal of Greenhouse Gas Control 39 (2015) 194–204

Author Response

REVIEWER #1

The contribution Membrane processes and carbon capture: State of the art & prospects gives a very good, comprehensive and at the same time concise overview on the use of membrane technology for carbon capture. It highlights the important aspects and most importantly gives good hints for the direction of future research.

The author thanks the reviewer for these encouraging comments.

There are a few minor issues, that should be addressed:

Page 1, line 12: Facilitated transport membranes should also be mentioned as an example

This is a good idea. It has been added.

Page 2, lines 62-64: This should be explained more clearly. Heat energy is worth less than mechanical energy?

Yes. Power (electricity) is generally considered as a high quality energy, while heat is a low one. A sentence has been added to detail this point.

Page 3, lines 78-84: It should be mentioned already here (later it is discussed) that generally water vapour up to saturation level is present and that various, potentially harmful trace components SO2, NOX should be considered when designing a unit.

This has been added.

Page 3, Table 1, “Steel row”: comma missing between H2 and N2

The change is made.

Page 5, line 134: Was this number calculated using the 2.7 factor mentioned above?

Yes. This information has been added in the caption.

Page 5, line 154: The large footprint area that would result when operating vacuum machinery at pressures of 10  to 100 mbar should be mentioned.

This point is added.

Page 7, line 195: please give a reference for these studies.

Two references have been added.

Page 7, Figure 5: (b) is missing in the figure caption.

The correction has been made.

Page 7, lines 209-214: One remark is that costs widely vary from country to country and even within one country as a function of time - just consider the current situation with Russia's invasion of the Ukraine and the resulting turmoil in the energy markets - maybe using power and duty requirements as outlined above is the fairer comparison.

This is a very relevant comment and it has been added.

Page 10, lines 297-305: Concentration polarisation an pressure drops as well as different membrane module designs were e.g. considered in International Journal of Greenhouse Gas Control 39 (2015) 194–204

The author thanks the reviewer. The reference has been added.

Reviewer 2 Report

The manuscript in general gives a nice overview on membrane materials and their performance, also presenting new viewpoint on the possibilities. The manuscript is submitted as an article, but at first sight, it mainly seems to collect the results of others and therefore gives the impression of a review rather than a research article. Some interesting results are however offered, but I recommend to further highlight the novelty of the results and the aims of the study in the introduction section. The results somewhat advance knowledge on the possibilities of membrane technologies and different materials, but even more focus on raising relevant questions for future research.

The author is correct in saying that there is a need for further research on membrane technologies used for flue gas CO2 capture. Membranes are however more commonly used in biogas plants than in flue gas applications, and it would be interesting to know more about why that is (feed gas properties like shown in Table 1, but is there more behind it?). Now the author merely states that membrane technologies should be promoted for flue gas use, without much addressing to their current uses, which could be improved.

Table 1 presents the main CO2 sources but lacks biomass boilers (power and heat plants, recovery boilers in kraft pulp mills, etc.). Biomass combustion is often neglected when carbon capture is discussed, likely because it is not incentivized. However, biomass boilers produce large amounts of CO2 and moreover, can even be a cheaper choice for carbon capture that smaller fossil sources.

Figure 3 could be informative but was not very clear. It could be clarified and opened up a bit more in the text, e.g., do the colors indicate something? The same applies to Figure 4: it would be easier to read if for example the lines with different xin values differed from each other (dashed or colored lines).

Figure 5: part (b) missing from the caption.

Figure 7: readability is poor, larger font size might help.

Author Response

REVIEWER #2

The manuscript in general gives a nice overview on membrane materials and their performance, also presenting new viewpoint on the possibilities. The manuscript is submitted as an article, but at first sight, it mainly seems to collect the results of others and therefore gives the impression of a review rather than a research article. Some interesting results are however offered, but I recommend to further highlight the novelty of the results and the aims of the study in the introduction section. The results somewhat advance knowledge on the possibilities of membrane technologies and different materials, but even more focus on raising relevant questions for future research.

The author thanks the reviewer for these comments. The paper is in fact a Perspective paper and not a classical research article. The title has been changed to highlight the key target of the paper, namely bridge the gap between materials and processes.

The author is correct in saying that there is a need for further research on membrane technologies used for flue gas CO2 capture. Membranes are however more commonly used in biogas plants than in flue gas applications, and it would be interesting to know more about why that is (feed gas properties like shown in Table 1, but is there more behind it?). Now the author merely states that membrane technologies should be promoted for flue gas use, without much addressing to their current uses, which could be improved.

This is an interesting point. A paragraph has been added in order to explain why membranes are considered as the best technology for biogas and not for carbon capture.

Table 1 presents the main CO2 sources but lacks biomass boilers (power and heat plants, recovery boilers in kraft pulp mills, etc.). Biomass combustion is often neglected when carbon capture is discussed, likely because it is not incentivized. However, biomass boilers produce large amounts of CO2 and moreover, can even be a cheaper choice for carbon capture that smaller fossil sources.

This is a relevant suggestion and it has been added.

Figure 3 could be informative but was not very clear. It could be clarified and opened up a bit more in the text, e.g., do the colors indicate something? The same applies to Figure 4: it would be easier to read if for example the lines with different xin values differed from each other (dashed or colored lines).

Figures 3 and 4 have been modified in order to improve clarity.

Figure 5: part (b) missing from the caption.

This has been added.

Figure 7: readability is poor, larger font size might help.

Fig 7 has been modified and enlarged.

Reviewer 3 Report

COMMENTS TO THE MANUSCRIPT  MEMBRANES-1907888

The manuscript tilted "Membrane processes and carbon capture : State of the art & prospects" presents a state of the art analysis, the future challenges and open questions of membranes for CO2 capture.

From my opinion, the scope and the content of the paper is of interest for Membranes, and my recommendation is Minor Revision, some points would be included to show more precisely the focus of the analysis, because the state of the art, challenges and open questions given here only focus on the membrane gas separations, and the CO2/N2 system, not in membrane contactors, not in the CO2 capture in other gas mixtures.

Let me indicate some comments to support my recommendation:

(1) The title is very general, but as I show the state of the art, challenges and open questions are focused to the membrane gas separations, and the CO2/N2 system, not in membrane contactors, not in the CO2 capture in other gas mixtures.

(2) Related to membrane materials, it is written in the abstract section "The possibilities offered by high performance membrane materials (zeolites, CMS, MOF, graphenes...) are discussed in combination to process strategies (multistage design, hybrid processes, energy integration)",

but section 3. Membrane materials is very short, and it is written there "A detailed analysis of the different performance levels that have been achieved for each type of material and separation mechanism is beyond the scope of this paper", and a synopsis in proposed in Table 2, and Figure 3 with a Robeson plot.

Some more discussion of this subject would be of interest to be included.

(3)  Related to the process, Section 4. Process Engineering, as single or multistage processes, it is related to membrane separations based on the component permeation (mainly CO2 / N2 systems as flue gas), it does not cover the CO2 capture by using membrane contactors, or other components.

Some discussion to cover more membrane technologies and CO2 capture from different sources a would be also of interest to be included.

 (4) As I think, Figure 7 should include the reference from it was adapted if is the case. The same comment is mentioned for the rest of figures.

Author Response

REVIEWER #3

The manuscript tilted "Membrane processes and carbon capture : State of the art & prospects" presents a state of the art analysis, the future challenges and open questions of membranes for CO2 capture.

From my opinion, the scope and the content of the paper is of interest for Membranes, and my recommendation is Minor Revision, some points would be included to show more precisely the focus of the analysis, because the state of the art, challenges and open questions given here only focus on the membrane gas separations, and the CO2/N2 system, not in membrane contactors, not in the CO2 capture in other gas mixtures.

The author thanks the reviewer for these comments.

Let me indicate some comments to support my recommendation:

(1) The title is very general, but as I show the state of the art, challenges and open questions are focused to the membrane gas separations, and the CO2/N2 system, not in membrane contactors, not in the CO2 capture in other gas mixtures.

The reviewer is right. The title has been changed to highlight that gas separation membranes only are discussed and focussed on post-combustion. This change should help the reader to catch that CO2/N2 separation only is analysed and no membrane contactor is detailed.

(2) Related to membrane materials, it is written in the abstract section "The possibilities offered by high performance membrane materials (zeolites, CMS, MOF, graphenes...) are discussed in combination to process strategies (multistage design, hybrid processes, energy integration)",

but section 3. Membrane materials is very short, and it is written there "A detailed analysis of the different performance levels that have been achieved for each type of material and separation mechanism is beyond the scope of this paper", and a synopsis in proposed in Table 2, and Figure 3 with a Robeson plot.

Some more discussion of this subject would be of interest to be included.

It is right that the materials section is (very) short and no detailed analysis is provided on the different materials categories. A small paragraph has been added, with comments on the different types.

(3)  Related to the process, Section 4. Process Engineering, as single or multistage processes, it is related to membrane separations based on the component permeation (mainly CO2 / N2 systems as flue gas), it does not cover the CO2 capture by using membrane contactors, or other components.

A sentence has been added on the process intensification part in order to mention membrane contactors.

Some discussion to cover more membrane technologies and CO2 capture from different sources a would be also of interest to be included.

A small paragraph has been added.

 (4) As I think, Figure 7 should include the reference from it was adapted if is the case. The same comment is mentioned for the rest of figures.

This is right. The references have been added in the captions.